# Association of Telomere Length in T Lymphocytes, B Lymphocytes, NK Cells and Monocytes with Different Forms of Age-Related Macular Degeneration

**DOI:** 10.3390/biomedicines12081893

**Published:** 2024-08-19

**Authors:** Anait S. Khalatyan, Anastasiya N. Shishparenok, Konstantin S. Avetisov, Yulia A. Gladilina, Varvara G. Blinova, Dmitry D. Zhdanov

**Affiliations:** 1Krasnov Research Institute of Eye Diseases, 11A, B, Rossolimo Str., Moscow 119021, Russia; avetisovks@gmail.com; 2Laboratory of Medical Biotechnology, Institute of Biomedical Chemistry, 10/8 Pogodinskaya St., Moscow 119121, Russia; a.shishparyonok@ibmc.msk.ru (A.N.S.); gladilinaya@ibmc.msk.ru (Y.A.G.); varya.blinova@list.ru (V.G.B.); zhdanovdd@gmail.com (D.D.Z.)

**Keywords:** age-related macular degeneration, macular atrophy, telomere length, lymphocytes, monocytes

## Abstract

Background: Age plays a primary role in the development of age-related macular degeneration (AMD). Telomere length (TL) is one of the most relevant biomarkers of aging. In our study, we aimed to determine the association of TL with T lymphocytes, B lymphocytes, NK cells or monocytes with different forms of AMD. Methods: Our study included 62 patients with AMD: geographic atrophy (GA), neovascular AMD (NVAMD) with and without macular atrophy and 22 healthy controls. Each leukocyte subtype was isolated from peripheral blood by immunomagnetic separation, and the DNA was purified. The TL in the genomic DNA was determined using qPCR by amplifying the telomere region with specific oligonucleotide primers and normalizing to the control gene. Statistical analysis was performed using R version 4.5.1. Results: We observed a statistically significant increase in TL in the T cells between the control and NVAMD groups but not for the GA group. The B cells and monocytes showed a significant decrease in TL in all AMD groups. The TL in the NK cells did not decrease in any of the AMD groups. Conclusions: The TL in the monocytes had the strongest association with AMD. It reflects a person’s “telomeric status” and may become a diagnostic hallmark of these degenerative processes.

## 1. Introduction

Telomeres are protective caps located at the 3′ ends of chromosomes and consist of a repeated (5′-TTAGGG-3′)n nucleotide sequence [1]. Most human somatic cells lose telomeres with each cell division due to incomplete lagging strand DNA synthesis during replication [2]. Depending on the cell type, tissue and method of measurement, the length of human telomeres has been found to be in the range of 1–15 kb [3,4,5,6]. Constant telomere attrition associated with cellular proliferation is a marker of biological aging [7,8,9], and shortened telomeres are a feature of some age-related diseases [10]. Telomere length (TL) has been linked to the onset and progression of Alzheimer’s disease [11], atherosclerosis and related vascular complications [12,13], coronary heart disease [14], abnormal reconstruction of bones [15] and different aspects of metabolic syndrome [16,17,18]. The majority of studies showing an association between TL and any disease have been performed on leukocytes, measuring the leukocyte TL (LTL). These cells, unlike those from soft tissues, are readily available and convenient to handle.

The telomere-associated theory of aging is reflected in ophthalmology, for instance, in neurodegenerative diseases, such as glaucoma and age-related macular degeneration (AMD) [19,20].

AMD is a chronic, progressive disease which affects elderly patients. AMD is divided into two forms: non-exudative (or “dry”) AMD and exudative (or “wet”) or neovascular (NVAMD) [21]. If left untreated, the advanced stages of this condition can result in the irreversible loss of central vision. The pathogenesis of AMD is currently being studied. It consists of multiple stages. Both forms of the disease begin with changes in the retinal pigment epithelium (RPE), with (1) the formation of drusen, leading to (2) drusenoid pigment epithelial detachments (PEDs). The terminal stage of the non-exudative form is (3a) geographic atrophy (GA) of the RPE and secondary photoreceptor atrophy. The advanced stage of exudative AMD refers to (3b) the development of macular neovascularization. Macular atrophy, the primary cause of severe central vision loss, can result from the advanced stages (3a,b) of both forms of AMD.

As the prevalence of AMD increases globally, it has become a significant non-infectious epidemic, posing severe public health, social and economic issues [22]. The etiology of AMD remains unknown, although there are several risk factors including age, smoking, ethnicity and family history [23]. However, there is increasing evidence supporting the genetic determinacy of AMD associated with telomere shortening [24,25].

In 2013, a study conducted by Ilkka Immonen and his colleagues revealed no significant link between the telomere length and AMD. The telomere length was analyzed in human DNA samples extracted from venous blood. Patients were unequivocally divided into groups, namely those with exudative AMD (83%) and those with non-exudative AMD, which included large drusen (14%) or central geographic atrophy (3%). The control group consisted of 77 subjects without AMD, except for patients who had only hard drusen <63 μ or minimal pigmentary abnormalities. The average telomere length in the patients with AMD and in the controls was not statistically significant [26]. However, a study conducted by Xiaoling Weng and his team in 2015 worked with a mixture of peripheral blood leukocytes and suggested a strong association between the leukocyte telomere length and AMD. Based on a sample size of 197 AMD patients and 259 healthy controls, the results showed that individuals with shorter telomeres had 2.24 times higher odds of having AMD [27]. Furthermore, this study found a highly significant association between geographic atrophy and shorter telomeres and no association with exudative AMD. Alvita Vilkeviciute et al. found that patients with early age-related macular degeneration (AMD) had longer telomeres in their blood samples compared with a control group. The researchers found that two variants, TRF2 rs251796 and TRF1 rs1545827, were linked to shorter telomeres. On the other hand, TRF1 rs10107605 was associated with reduced chances of developing early and exudative AMD. They also discovered that the TERF1 serum levels were higher in people with early AMD than those in the control group [28].

To date, the physiological and molecular relationships between the telomere length in peripheral blood leukocytes and types of AMD remain to be determined. All studies determining LTL in AMD patients only describe the phenomenon of telomere loss. Furthermore, a major limitation of the above studies is that the results were obtained using peripheral blood mononuclear cells (PBMCs), which are a mixture of T cells, B cells, NK cells, monocytes and dendritic cells. PBMCs are commonly obtained from buffy coats after density gradient centrifugation on separation media, as described elsewhere [29,30]. Different subpopulations of leukocytes participate in pathophysiological processes in different ways and have different proliferative and functional activities [31,32,33,34], suggesting their heterogeneity in telomere length. The measurement of LTL in a sample of PBMCs cannot provide sufficient information about the most representative cell type of leukocytes for attribution to AMD.

In our work, we aim to determine the TL in T lymphocytes, B lymphocytes, NK cells or monocytes from peripheral blood of AMD patients to better match with the different forms of this age-related disease.

## 2. Materials and Methods

### 2.1. Demographic Characteristics of Study Participants

A total of 84 patients (168 eyes) were examined and treated at the Department of Retina and Nerve Pathology of the Krasnov Research Institute of Eye Diseases. This study was approved by the Ethics Committee of the Krasnov Research Institute of Eye Diseases (Protocol No. 86/2 on 14 November 2022) and complied with the ethical principles of the Declaration of Helsinki. Informed consent was obtained from all patients.

The average age of the patients was 79 ± 9 years. Patients were not divided into groups by gender or age. All patients underwent a standard ophthalmological examination along with special diagnostic methods, including OCT and OCTA of the macular area using the Spectralis HRA + OCT (Heidelberg Engineering, Heidelberg, Germany). The patients were divided into four groups: a control group (*n* = 22), a group with end-stage non-exudative AMD (known as geographic atrophy (GA); *n* = 21) [35], a group of patients with neovascular AMD (NVAMD) with macular atrophy (MA; *n* = 20) [36], and a group of patients with neovascular AMD without MA *(n* = 21) [37]. The control group included individuals who did not have a diagnosis of AMD or early changes in the RPE by the age of 70, which means the complete absence of drusen as per optical coherence tomography examination.

The inclusion criteria for the group with end-stage non-exudative AMD were the absence of a history of antiangiogenic therapy, the absence of macular neovascularization and the presence of confluent, clearly defined foci of atrophy of the pigment epithelium, neuroepithelium and choriocapillaris layers according to optical coherence tomography. Patients in the latter two groups received anti-VEGF therapy for neovascular AMD (active macular neovascularization) and had a history of at least 20 injections in one eye. The inclusion criteria for these two groups were a BCVA of at least 20/40 Snellen (≤0.3 LogMAR) and conserved central vision. MA was defined as RPE degeneration accompanied by corresponding signal hypertransmission into the choroid and degeneration of the overlying photoreceptors.

Patients undergoing antiangiogenic therapy received intravitreal injections of 0.05 mL (2 mg) of aflibercept in an operating room using a 27 G needle, with at least 3.5 mm from the limbus, following standard procedures. Three injections were performed at four-week intervals, followed by a treat-and-extend regimen.

The exclusion criteria for all groups in this study included the presence of any acute eye disease, glaucoma, diabetic retinopathy, myopic maculopathy, uveitis of any cause, mature complicated cataracts, retinal detachment, rubeosis iridis and pathology of the vitreomacular interface with a traction component. Additionally, patients with autoimmune and oncological conditions in any part of the body were not included in this study.

### 2.2. Isolation of Leukocytes and Monocytes from Peripheral Blood

Peripheral blood at a volume of 16 mL was collected by venipuncture in Vacuette vacuum tubes with K3EDTA anticoagulant (Greiner Bio-One, Kremsmünster, Austria). Density gradient centrifugation of the blood using Ficoll (1.077 g/mL, Paneco, Moscow, Russia) as the separation medium was performed to obtain peripheral blood mononuclear cells (PBMCs). Single cells were isolated from the PBMCs through immunomagnetic labeling in MS MiniMACS separation columns (#130-042-201, Miltenyi Biotec, Bergisch Gladbach, Germany). The following microbeads were used for selective isolation of cells (all from Miltenyi Biotec, Bergisch Gladbach, Germany) according to the manufacturer’s protocol. Human CD3 MicroBeads (#130-050-101) were used for the T lymphocytes; human CD19 MicroBeads (#130-050-301) were used for the B lymphocytes; human CD56 MicroBeads (#130-050-401) were used for the NK cells; and human CD14 MicroBeads (#130-050-201) were used to isolate the monocytes. The purity of the isolated cells was monitored with a flow cytometry MACSQuant Analyzer 10 (Miltenyi Biotec, Bergisch Gladbach, Germany) using a human 7-Color Immunophenotyping Kit (#130-098-456). DNA was isolated from each cell using a DU-250 DNA Extraction Kit (Biolabmix, Novosibirsk, Russia). The integrity of the isolated DNA was visualized using 1% agarose gel electrophoresis in TAE buffer stained with ethidium bromide. The concentration of the isolated DNA was measured via fluorometric quantification using a Qubit 4 fluorometer (Thermo Fisher Scientific Inc., Waltham, MA, USA). The samples and DNA were stored at −80 °C prior to analysis.

### 2.3. Analysis of Leukocyte Telomere Length by Quantitative PCR (qPCR)

A telomere length in the genomic DNA samples was determined through qPCR by amplifying the telomere region with specific oligonucleotide primers according to previously described reference methods [38,39,40]. The ratio of telomere repeat copies to the quantitative control gene was estimated. For quantitative control (endogenous control of the DNA load) and to determine the genome copies per sample, the ribosomal phosphoprotein P0 (36B4) gene was used as an invariable single-copy gene. The specific primers for the telomere region were as follows: forward 5′-ggtttttgagggtgagggtgagggtgagggtgaggt-3′ and reverse 5′-tcccgactatccctatccctatccctatccctatccctatcccta-3′. The specific primers for 36B4 were as follows: forward 5′-cagcaagtgggaaggtgtaatcc-3′ and reverse 5′-cccattctatcatcaacgggtacaa-3′. Primers were custom-synthesized by Evrogen (Moscow, Russia). PCR was performed using qPCR mix-HS SYBR (Evrogen, Moscow, Russia) with 20 ng genomic DNA and 200 nM telomere or 36B4 pair primers in a DTprime 5 Real-Time PCR System (DNA-Technologies, Protvino, Russia). The thermal cycling profile for both amplicons was as follows: 95 °C for 3 min for activation of the HS polymerase, followed by 40 cycles of 20 s at 95 °C for melting, 40 s at 60 °C for primer annealing and 30 s at 72 °C for extension. Each sample was run in triplicate. The sample without DNA was added to each run as a negative control for cross-contamination. A cycle threshold (Ct) was determined for each sample, which is the number of cycles at which the fluorescence signal from the amplicons exceeds the set detection threshold. The threshold was set to be the same for all samples. Amplification results were considered acceptable if the coefficient of variation (CV) of three Ct values was less than 2% for both amplicons.

To measure the amount of telomeric sequence per sample (in kb), the Ct of the samples was compared to standard calibration curves for telomere and 36B4 [38]. To create them, oligomer telomer standard (5′-ttaggg-3′)14 or 36B4 standard 5′-cagcaagtgggaaggtgtaatccgtctccacagacaaggccaggactcgtttgtacccgttgatgatagaatggg-3′ (both custom-synthesized by Syntol, Moscow, Russia) were amplified under the conditions described above. Plasmid DNA pTagGFP2-N (Evrogen, Moscow, Russia) was added to each sample to maintain a constant amount of 20 ng DNA per reaction. The number of telomere sequence repeats was 1.18 × 10^8^ kb for the telomere standard oligonucleotide and 2.63 × 10^9^ genome copies per reaction for the 36B4 standard oligonucleotide at the highest concentrations in the reaction mixture. These numbers were calculated with the standard techniques described by O’Callaghan [38]. A serial dilution from 10-1 to 10-6 of telomere or 36B4 standard oligonucleotides was subjected to qPCR. The value of kb/reaction was then used to determine the total telomere length per human diploid genome. For this, the telomere kilobase count was divided by the diploid genome copy number to obtain the total telomere length in kilobases per human diploid genome per sample. This value was then divided by 92, the total number of telomeres on 23 pairs of chromosomes, to obtain a mean telomere length expressed in kilobases per sample.

### 2.4. Statistics

Statistical analysis was performed using R version 4.5.1 (R Foundation for Statistical Computing, Vienna, Austria; URL: https://www.R-project.org/, accessed on 5 May 2024). Descriptive data for the leukocyte and monocyte telomere lengths within each group were presented as a median with an interquartile range (IQR). The normality of the data distribution was checked using the Shapiro–Wilk test. The Mann–Whitney U test was used to compare the telomere lengths for each pair of groups. Bonferroni correction was used to account for multiple comparisons. Analysis included 6 pairs of comparison groups for each peripheral blood parameter. Therefore, differences were considered statistically significant when *p* < 0.008 (0.05/6 = 0.008).

## 3. Results

### 3.1. Cell Purity, DNA Integrity and Construction of Standard Curves

Different cell types even from the same organism may have different TLs [3,4,5,6]. Therefore, the subpopulations of isolated leukocytes must be homogeneous. Using immunomagnetic separation, we obtained T cells, B cells, NK cells or monocytes from the PBMCs with a purity of more than 90% (Figure 1A–D, respectively). The integrity of the isolated DNA is of primary importance, as DNA degradation decreases the detected TL. Using agarose gel electrophoresis, we demonstrated that the isolated DNA from the leukocytes was not degraded (Figure 1E).

The quality of reagents and qPCR instrumentation can affect the results of a TL measurement. The construction of calibration curves using the oligomer telomere standard and 36B4 standard reflects the quality of the technique used. We obtained standard curves with coefficients of determination (R2) greater than 0.99 for both the telomere standard (Figure 2A) and the reference standard (Figure 2B). The high quality of the calibration curves indicates that the techniques used are applicable to the detection of TLs in patient DNA samples.

### 3.2. TLs in Different Subpopulations of Leukocytes from Control Patients

The first observation of our study is the difference in the TLs of the leukocyte subpopulations from the control group (Table 1, Figure 3). Monocytes and B cells had the longest telomeres (9.46 ± 1.60 kb and 8.84 ± 2.26 kb, respectively), whereas T cells had the shortest telomeres (3.97 ± 1.73 kb). The TL in the NK cells was rather moderate (5.70 ± 2.03 kb). The comparison of telomere lengths among leukocyte subpopulations from the control patients is shown in Appendix A.

### 3.3. TLs in Leukocytes from Patients with Different Types of AMD

We found statistically significant differences in the TLs of subpopulations of leukocytes between the control group and each of the AMD groups (Table 1, Figure 3). The TL in the T cells (Figure 3A) was increased up to 8.26 ± 2.59 kb in the NVAMD MA group and up to 7.79 ± 2.34 kb in the NVAMD no MA group. The TL of the T cells from the GA group (6.68 ± 2.16) did not differ from that of the control group, but a statistical difference was found between the GA and NVAMD groups. In Appendix A, we have provided the *p* values for paired comparisons of the TLs of leukocytes from each AMD group.

The TL in the B cells was significantly reduced compared with the control in all groups of AMD patients (Figure 3B), being 4.82 ± 2.36 kb in the NVAMD MA group, 5.74 ± 2.84 kb in the NVAMD no MA group and 5.20 ± 2.34 kb in the GA group. The difference in the TLs of the B cells between AMD groups was not significant.

The TL in the NK cells compared with the control or within AMD groups showed no significant statistical difference (Figure 3C). The TL was 6.30 ± 2.87 kb in the NVAMD MA group, 6.97 ± 2.97 kb in the NVAMD no MA group and 6.68 ± 2.16 kb in the GA group.

Monocytes showed the most significant TL shortening in the AMD groups compared with the control (Figure 3D), being 3.99 ± 2.55 kb in the NVAMD MA group, 4.54 ± 2.36 kb in the NVAMD no MA group and 5.23 ± 2.15 kb in the GA group. As in the case of the B cells, we found no statistical difference in the TLs of monocytes between AMD groups.

To assess which leukocyte was associated with the greatest difference between TLs in the control and AMD groups, we increased the precision of the *p* values in Appendix A. As a result, we found that in all cases, the TLs of the monocytes in the control group deviated more from the TLs of the monocytes in the NVAMD, NVAMD no MA or GA groups compared with other leukocytes (Appendix A). Thus, we can conclude that the longest telomeres of monocytes in the control group showed the most dramatic loss in AMD patients, and these cells were the most representative ones for association with this age-related disease.

## 4. Discussion

According to the concept of cellular senescence, telomeres and telomere-associated proteins play a critical role in the aging process. Telomere shortening is associated with metabolic, inflammatory and neurodegenerative diseases directly related to aging [41]. Age is the main risk factor for AMD because RPE cells are affected by aging and degenerative processes, and their replicative capacity decreases with age [10]. Aging RPE cells are susceptible to oxidative stress [42], and they experience accelerated telomere shortening as a result of repeated cell division [43]. Induction of telomerase, a multienzyme capable of extending telomeres, results in maintenance of the characteristics of early passage cells in terms of protein expression, cell cycle distribution and melanin formation [44].

Many studies have shown that the TL in white blood cells is associated with the onset and progression of AMD [20,27,28,45,46]. However, there is no clear evidence of the direct involvement of AMD progression in the biology of telomeres of cells in the peripheral blood.

The main limitation of our study is the small size of the control group (*n* = 22) and the AMD group (*n* = 20–21). However, the size of the groups was sufficient to demonstrate statistical significance between them. Careful selection of patients according to inclusion criteria, high purity of cells after immunomagnetic separation, standardization of isolated DNA as well as selection of reagents and refinement of the telomere length detection procedure were necessary to obtain clear results.

In our study, we observed a difference in the TLs of leukocyte subpopulations from the control group of donors (Table 1, Figure 3). Such a difference may be explained by individual proliferative activity sensitivity for stimulation between these cells. T, B and NK lymphocytes participate in multiple immunological processes and undergo many stages of activation (i.e., antigenic stimulation) [47,48] and suppression [49]. Upon activation, lymphocytes retain the ability to rapidly produce a large number of cells. Immune activation of these cells is usually accompanied by the induction of telomerase [50,51,52], while the reduction in telomerase activity is attributed to the suppressive immune processes [53,54]. Therefore, two opposing dynamic processes of telomere loss upon proliferation and telomere lengthening during telomerase activation determine the total individual TL in lymphocytes.

The unexpected observation from our results is that initially, the shortest TL in the T cells increased rather significantly in the NVAMD MA and NVAMD no MA groups (Table 1, Figure 3A). We can explain this phenomenon with the strong association of T cells with degenerative diseases [55], including AMD [56]. The breakdown of ocular immune tolerance involves the blood-retinal barrier and anti-inflammatory and anti-immune proteins, leading to a targeted attack by effector T cells on autoantigens [57,58]. Prolonged activation and autoactivation of T cells lead to the induction of telomerase and abnormal telomere elongation.

The TL in the B cells was significantly shorter in all three groups of AMD patients (Table 1, Figure 3B). This observation is in accordance with previous studies, which demonstrated a decrease in TL in PBMCs from AMD patients [20,27,28,45,46]. The relationship between B lymphocytes and AMD is not clear, and more research is needed. The number of B lymphocytes changes with age, possibly due to increased autoimmunity in the elderly. However, there are no significant differences in AMD compared with healthy individuals [59].

The TL in the NK cells was unchanged in the AMD groups compared with the controls (Table 1, Figure 3C). NK cells were initially thought to play a role in developing AMD when Khan et al. performed a genotyping study [60]. The study demonstrated an association between the HLA-Cw*0701 allele and killer cell immunoglobulin-like receptor ligand haplotype AA with AMD [60,61]. However, in a recent gene expression study, researchers found a lower prevalence of resting NK cells in AMD patients, but they observed a positive correlation between the C1S, ADM and 1ER5L genes and NK cell activation, as well as AMD progression [62]. Obviously, these processes are unlikely to affect the TL in peripheral NK cells.

Monocytes were the most representative peripheral blood cells associated with AMD (Table 1, Figure 3D). The most likely explanation for this is that unlike lymphocytes, mature monocytes do not express telomerase and do not undergo further cell division after activation [63]. We can assume that the decrease in TL in monocytes with age likely reflects the decrease in TL in hematopoietic progenitor cells. There are several potential molecular mechanisms which may affect the TL in monocytes with aging [64]. A universal mechanism associated with a broad spectrum of intracellular damage during aging is reactive oxygen species. Exposure to them leads to cross-linking of mitochondrial and intracellular macromolecules which impair their function. Another likely cause of monocyte aging is the defects in genomic DNA and the reduced ability to repair DNA damage.

Different dynamics (increasing T cells and decreasing B cells or monocytes) of the TLs between cells in the AMD and control groups is the most likely reason for the weak correlation [27,28] or even lack of a difference [26] in previous studies performed on PBMCs. In such experiments, the specific fraction of T cells, which are in the majority in PBMCs and have elongated telomeres, will increase the mean value of the TL in the measured sample. The measured TLs of minor B cells and monocytes with shortened telomeres could slightly lower the mean TL value in the PBMC sample.

The retina is protected from the bloodstream by a blood-retinal barrier, preventing uncontrolled substance diffusion between blood and the retina. The loss of barrier integrity is directly linked to neovascularization, stemming from the absence of the characteristic blood–brain barrier structure in the new vessels observed in AMD [65].

The different TL in leukocyte subpopulations raises the question of determining the frequencies of T, B and NK cells or monocytes in AMD patients. Several studies have investigated the frequency of monocytes in patients with AMD. Lechner et al. [66] investigated the relationship between the frequency of circulating white blood cell populations and the prevalence and clinical manifestations in nAMD patients. The percentage of circulating CD11b^+^ cells and CD16^hi^HLA-DR^−^ neutrophils was significantly higher in the exudative AMD patients compared with the controls. A higher percentage of CD4^+^ T cells was found in patients with subretinal fibrosis compared with those without. Changes in the leukocyte profile of AMD patients, especially with regard to monocytes, have been reported in other studies [67,68]. In a study by Hector S.M. et al., the percentage of total monocytes was lower in the nAMD group compared with the control group [59]. The percentage of peripheral blood CD163^+^ monocytes was higher in both the non-exudative and exudative AMD patients compared with the age-matched non-AMD controls. However, there was no difference in the percentages of peripheral CD206^+^ and CD80^+^ monocytes between the groups [69].

## 5. Conclusions

It seems quite unlikely that the development of AMD is a primary cause of changes in monocyte TLs. It is also unlikely that abnormal TLs in these cells are the primary cause of AMD. Therefore, we can speculate that the relationship between the TLs in monocytes of different types of AMD is highly indirect. According to our results, the strongest association of AMD types was with the TL in monocytes rather than in other leukocytes. The TL in monocytes reflects a person’s “telomeric status” and may become a diagnostic hallmark of these degenerative processes. The physiological and molecular links between the TL in monocytes and types of AMD remain to be determined.

## Figures and Tables

**Figure 1 biomedicines-12-01893-f001:**
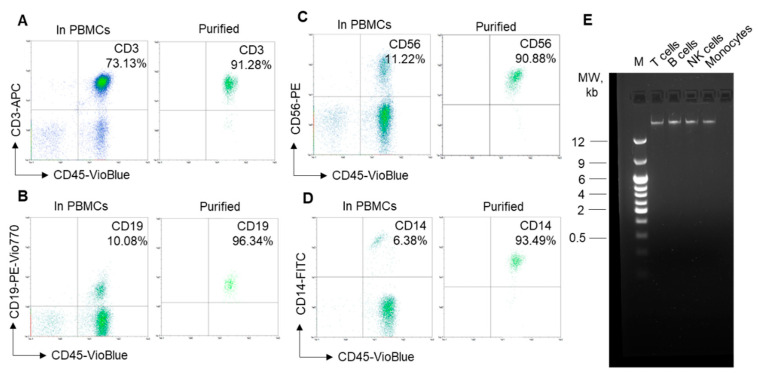
High purity of separated leukocyte subpopulations and the integrity of the isolated DNA. (**A**) Representative flow cytometry plots showing the proportion of leukocyte subpopulations before separation (in PBMCs) and after immunomagnetic separation (purified cells). (**A**) T lymphocytes labeled with CD45-VioBlue and CD3-APC antibodies. (**B**) B lymphocytes labeled with CD45-VioBlue and CD19-Vio770 antibodies. (**C**) NK cells labeled with CD45-VioBlue and CD56-phycoerythrin (PE) antibodies. (**D**) Monocytes labeled with CD45-VioBlue and CD14-FITC antibodies. (**E**) Agarose gel electrophoresis of isolated DNA from leukocyte subpopulations demonstrating DNA integrity. M = molecular weight (MW) marker in kilobases.

**Figure 2 biomedicines-12-01893-f002:**
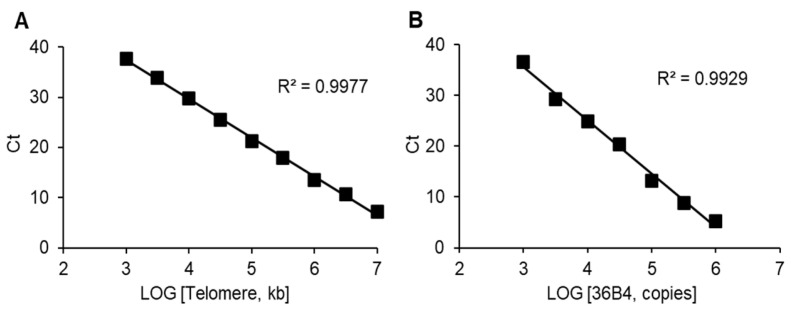
The standard curves used to calculate the lengths of telomere sequences per sample. (**A**) The dependence of the cycle threshold (Ct) on the concentration of the telomere standard (kb per reaction). (**B**) The dependence of the Ct on the concentration of 36B4 copy number. Linear ranges of the curves are shown. Linear Ct ranges were used for further calculations of telomere/control ratio.

**Figure 3 biomedicines-12-01893-f003:**
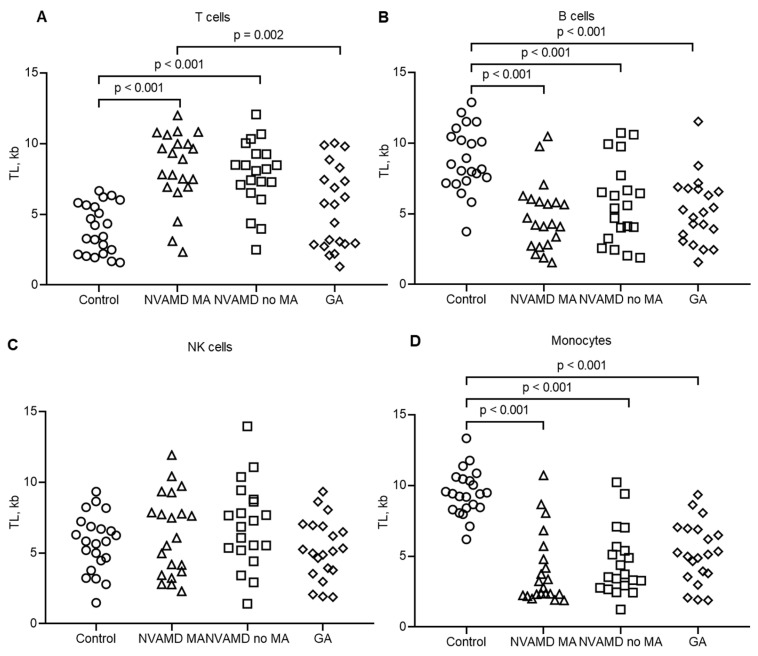
Individual TL in leukocytes form AMD patients and control donors. (**A**) T cells. (**B**) B cells. (**C**) NK cells. (**D**) monocytes. The significance of the difference (*p*-value) is shown.

**Table 1 biomedicines-12-01893-t001:** Descriptive information about leukocyte TLs in each group. Median and IQR are shown.

Leukocytes	Control (*n* = 22)	NVAMD MA (*n* = 21)	NVAMD No MA (*n* = 20)	GA (*n* = 21)
T cells	3.83 (2.28–5.62)	8.9 (6.94–9.98)	8.13 (6.94–9.26)	5.7 (2.9–7.44)
B cells	8.34 (7.39–10.39)	4.27 (2.82–5.85)	5.49 (3.83–6.92)	5.1 (3.54–6.75)
NK cells	5.83 (4.52–6.83)	6.08 (3.69–7.86)	7.07 (5.32–8.66)	6.44 (5.11–8.36)
Monocytes	9.41 (8.41–10.42)	2.77 (2.31–4.78)	3.62 (2.89–5.46)	5.13 (3.79–6.89)

## Data Availability

Data are available upon request via the corresponding author.

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
