# Peer review of "Association of Telomere Length in T Lymphocytes, B Lymphocytes, NK Cells and Monocytes with Different Forms of Age-Related Macular Degeneration"

_biomedicines, 2024, doi:10.3390/biomedicines12081893_

Round 1

Reviewer 1 Report

Comments and Suggestions for Authors

Dear authors:

Thanks for giving us a new concept about ARMD.

Thanks.    

Author Response

We appreciate the reviewer’s positive characterization of our manuscript.

We thank all reviewers for their comments and suggestions.

Reviewer 2 Report

Comments and Suggestions for Authors

Dear authors, It has been a pleasure to review the manuscript titled "Association of Telomere Length in T lymphocytes, B lymphocytes, NK cells and Monocytes with Different Forms of Age-Related Macular Degeneration". After thorough review, I regret to mention that the manuscript cannot be accepted for publication at this time.

Concerns have been identified related to the novelty and depth of the study. The study is lacking in novelty as there are previous studies that report the same results presented in this manuscript. The authors should have better highlighted and related the presence of shorter telomeres in different types of AMD. However, with the simplistic analysis conducted, no further conclusions can be drawn beyond what has already been presented. Additionally, there is a lack of experimental work that could have guided the results and conclusions in a more impactful direction.

Nevertheless, I have attached some comments that may assist the authors in enhancing the article.

-        Firstly, the title does not convey any novel findings from the study. The authors should consider reorienting the study to introduce novel insights, thereby achieving results with significant clinical and scientific relevance.

-        The manuscript should have line numbers for easier reference and review. This will help reviewers and readers to provide more precise feedback and comments on specific sections of the text.

-        The introduction in the abstract is very weak. It does not provide sufficient background or context, nor does it relate AMD to immune cells. The introduction should be expanded to clearly establish the connection between AMD and immune cells, providing a strong foundation for the study.

-        The conclusion of the abstract does not relate the results to the status of the different types of AMD. The authors should delve much deeper into this aspect and highlight the clinical implications of these results.

-        In the introduction, the authors should justify why the study is conducted in leukocytes.

-        The telomere-associated theory of aging is highly relevant in ophthalmology and should be developed in the manuscript due to its importance in this context. The authors should elaborate on this theory to provide a stronger theoretical framework for their study.

-        The pathogenesis stages described in the formation of AMD should be separated according to the different types of AMD. Currently, the description is unclear and does not distinguish between the distinct forms of AMD.

-        In the sentences "As the prevalence of AMD increases globally, it has become a significant non-infectious epidemic, posing severe public health, social, and economic issues. The etiology of AMD remains unknown, although there are several risk factors including age, smoking, ethnicity, and family history," there are no bibliographic references to support these claims.

-        When stating "However, there is increasing evidence supporting the genetic determination of the disease associated with telomere shortening," it is unclear what specific disease or condition is being referred to. Is this statement related to AMD specifically, or does it encompass other diseases in general? Additionally, the manuscript does not provide references distinct from those cited in previous paragraphs to support this claim.

-        Regarding the study by Ilkka Immomen et al., could you please specify where the telomeres were measured? Additionally, please clarify in which types of AMD and using which animal or human models were these measurements conducted.

-        In the study by Xiaoling Weng, the specific cells or tissues in which the telomere measurements were conducted are not mentioned. Please provide details on the cell types or tissues analyzed in this study to better understand the relevance of their findings in relation to your research.

-        A point of concern is the apparent contradiction between studies on the association between telomeres and AMD. For instance, Xiaoling Weng reports telomere shortening, while Alvita Vilkeviciute et al. observe telomere elongation in early stages of AMD, although the specific type of AMD is not mentioned. Could you clarify the reasons behind these conflicting findings and discuss their implications?

-        At the end of the introduction, there is no clear discussion of how this study differentiates from the previously mentioned studies, all of which use PBMCs. This lack of distinction and novelty among the studies is a significant concern. Additionally, the specific objective of this study is not clearly articulated in relation to the prior work. Clearly defining the study's objective is crucial, as it will guide the remainder of the work and must be novel to contribute significantly to scientific and clinical relevance.

-        The manuscript lacks details regarding the treatment regimen and the extent of antiangiogenic therapy administered to the patients.

-        I believe that the analysis conducted in the manuscript should be complemented with additional methods to verify the results. For instance, employing fluorescently labeled nucleic acid probes that specifically bind to telomeric repeats for Q-FISH could provide more detailed insights into telomere length by measuring fluorescence using flow cytometry. Alternatively, methods such as STELA could be used to detect shorter telomeres on specific chromosomes. Incorporating these techniques would strengthen the robustness of the findings and offer a more comprehensive assessment of telomere dynamics.

-        The Statistical Analysis section appears limited in detail. Could the authors expand significantly to provide a more thorough description of the statistical analyses employed, including the rationale behind the selection of methods? Additionally, it would be helpful to explain why a p-value of 0.008 was considered significant and to include details on effect size and estimated statistical power for the analyses performed.

-        To ensure the effectiveness and purity of the cellular separations, the authors should have included a comparison of the values obtained using gradient centrifugation versus immunomagnetic separation. Providing this comparison would help validate the purity of the cell populations used in the study and ensure that the results are reliable.

-        I am concerned that the current study does not present a significant novelty, as similar methodologies, such as magnetic separation of leukocytes for ocular diseases, have already been explored in previous publications. For instance, Vazirpanah et al. have conducted related research (https://journals.plos.org/plosone/article?id=10.1371/journal.pone.0176175). The study should emphasize how it differs from or builds upon these existing studies to demonstrate its unique contributions to the field.

-        I have noted inconsistencies between the p-values reported in the tables and those described in the Materials and Methods section. For instance, in Table S1, the p-value for NK cells versus T cells is reported as 0.006, while the Materials and Methods section states that a p-value of 0.008 is considered significant. Additionally, similar discrepancies are observed in Tables S2 and S5, where p-values are reported as being less than 0.008 but greater than 0.001. The authors should address these contradictions by clarifying the criteria used for significance and ensuring consistency across all reported values.

-        In Table 1, the term "IQR" is used but not defined.

-        The study would benefit from a discussion on the concept that the eye is an immunoprivileged organ, isolated from the rest of the body by the blood-retinal barrier. The authors should explain the relevance of studying peripheral blood cells in the context of an ocular disease that primarily affects the retina, which has this controlled barrier. How do the findings in peripheral blood cells relate to the retinal pathology of age-related macular degeneration, given the isolation and unique immune environment of the eye?

Author Response

We appreciate the reviewer’s responses. In the following response, we address each calling for changes, indicating where relevant corrections have been added to the body of the manuscript and their locations. For reviewers’ convenience, all corrections are tracked in red color in the main text file.

We thank all reviewers for their comments and suggestions.

Response to the comments from the Reviewer #2

Comment 1

Concerns have been identified related to the novelty and depth of the study. The study is lacking in novelty as there are previous studies that report the same results presented in this manuscript. The authors should have better highlighted and related the presence of shorter telomeres in different types of AMD. However, with the simplistic analysis conducted, no further conclusions can be drawn beyond what has already been presented. Additionally, there is a lack of experimental work that could have guided the results and conclusions in a more impactful direction.

Response:

To the best of our knowledge, all previous studies determining telomere length in leukocytes from AMD patients have used the peripheral blood mononuclear cell (PBMC) fraction obtained after gradient centrifugation. PBMCs consist of a mixture of white blood cells, which limits the interpretation of the results. In our work, we isolated individual subpopulations (i.e., T lymphocytes, B lymphocytes, NK cells, or monocytes) and demonstrated for the first time the association of telomere length with different types of AMD. This is the novelty of the paper. Such results have not been published before. All necessary experiments were performed to support our hypothesis.

All of the above were stated in the Introduction section.

Comment 2

Firstly, the title does not convey any novel findings from the study. The authors should consider reorienting the study to introduce novel insights, thereby achieving results with significant clinical and scientific relevance.

Response:
Again, the main finding of our work is that we have demonstrated for the first time the association of telomere length in T lymphocytes, B lymphocytes, NK cells and monocytes with different forms of age-related macular degeneration. This is the novelty of our paper and the novelty is reflected in the title.

Comment 3

The manuscript should have line numbers for easier reference and review. This will help reviewers and readers to provide more precise feedback and comments on specific sections of the text.

Response

Thank you for your concern. We have added the line numbers to the revised file.

Comment 4

The introduction in the abstract is very weak. It does not provide sufficient background or context, nor does it relate AMD to immune cells. The introduction should be expanded to clearly establish the connection between AMD and immune cells, providing a strong foundation for the study.

Response

To date, the physiological and molecular relationships between telomere length in peripheral blood leukocytes and types of AMD are not understood. All previous studies determining telomere length in leukocytes from AMD patients describe only the phenomenon of telomere loss. It seems very unlikely that the development of AMD is a primary cause of changes in leukocytes TL. It is also unlikely that abnormal TL in these cells is the primary cause of AMD. Therefore, we can assume that the relationship between leukocyte TL and different types of AMD is highly indirect and that telomere length reflects a person's "telomeric status". In our work, we aimed to determine TL in T lymphocytes, B lymphocytes, NK cells or monocytes from peripheral blood of AMD patients to better match with the different forms of AMD.

We have changed the introduction to reflect this concern and to emphasize the link between immune cells and AMD.

Comment 5

In the introduction, the authors should justify why the study is conducted in leukocytes.

Response

The study was performed on leukocytes because the results of the previous studies were obtained using the fraction of PBMCs, which is a mixture of white blood cells (mainly leukocytes). Measuring telomere length in a sample of PBMCs may not provide sufficient information about the most representative cell type of leukocytes for attribution to AMD. In our work, we aimed to determine telomere length in T lymphocytes, B lymphocytes, NK cells or monocytes from the peripheral blood of AMD patients to better correlate with the different forms of AMD.

All of the above were stated in the Introduction section.

Comment 6

The telomere-associated theory of aging is highly relevant in ophthalmology and should be developed in the manuscript due to its importance in this context. The authors should elaborate on this theory to provide a stronger theoretical framework for their study.

Response

We are very sorry, but the meaning of this concern is not clear to us. In the Introduction section, we stated that the telomere-associated theory of aging is reflected in ophthalmology, for example, in neurodegenerative diseases - glaucoma and age-related macular degeneration. The elaboration of this theory is more suitable for a review than for an experimental paper.

Comment 7

The pathogenesis stages described in the formation of AMD should be separated according to the different types of AMD. Currently, the description is unclear and does not distinguish between the distinct forms of AMD.

Response

The stages 1,2 refer to both non-exudative and exudative forms of AMD. Stage 3a refers to the terminal stage of the non-exudative form of AMD. Stage 3b refers to exudative AMD.

We have introduced the classification of AMD forms to the introduction section.

Comment 8

In the sentences "As the prevalence of AMD increases globally, it has become a significant non-infectious epidemic, posing severe public health, social, and economic issues. The etiology of AMD remains unknown, although there are several risk factors including age, smoking, ethnicity, and family history," there are no bibliographic references to support these claims.

Response

Thank you! The references are provided in the revised manuscript.

Comment 9

When stating "However, there is increasing evidence supporting the genetic determination of the disease associated with telomere shortening," it is unclear what specific disease or condition is being referred to. Is this statement related to AMD specifically, or does it encompass other diseases in general? Additionally, the manuscript does not provide references distinct from those cited in previous paragraphs to support this claim.

Response

This statement is related to AMD specifically. We provided the references in the revised manuscript. Thank you!

Comment 10

Regarding the study by Ilkka Immomen et al., could you please specify where the telomeres were measured? Additionally, please clarify in which types of AMD and using which animal or human models were these measurements conducted.

Response

Telomere length was analyzed in human DNA samples extracted from venous blood. Patients were unequivocally divided into groups: those with exudative AMD (83%), and those with non-exudative AMD, which included large drusen (14%) or central geographic atrophy (3%). The control group consisted of 77 subjects without AMD, except for patients who had only hard drusen <63μ or minimal pigmentary abnormalities.

This information has been added to the revised introduction

Comment 11

In the study by Xiaoling Weng, the specific cells or tissues in which the telomere measurements were conducted are not mentioned. Please provide details on the cell types or tissues analyzed in this study to better understand the relevance of their findings in relation to your research.

Response

The information about cell type has been added to revised manuscript. Thank you!

Comment 12

A point of concern is the apparent contradiction between studies on the association between telomeres and AMD. For instance, Xiaoling Weng reports telomere shortening, while Alvita Vilkeviciute et al. observe telomere elongation in early stages of AMD, although the specific type of AMD is not mentioned. Could you clarify the reasons behind these conflicting findings and discuss their implications?

Response

Thank you for your suggestion. This contradiction is exactly the reason why we decided to determine the telomere length in individual subsets of leukocytes. Moreover, this contradiction can be explained by the results of our work.

A major limitation of the above studies is that the results were obtained using peripheral blood mononuclear cells (PBMCs), which is a mixture of T cells, B cells, and NK cells, monocytes, and dendritic cells. PBMCs are commonly obtained from buffy coats after density gradient centrifugation on separation media. Different dynamics (increasing in T cells and decreasing in B cells or monocytes) of TL between cells in AMD and control groups is the most likely reason for the weak correlation [27,28] or even lack of difference [26] in previous studies performed on PBMC cells. In such experiments, the specific fraction of T cells, which are the majority in PBMCs and have elongated telomeres, will increase the mean value of TL in the measured sample. The measured TL of minor B cells and monocytes with shortened telomeres could slightly lower the mean TL value in the PBMC sample.

We have discussed this issue in our introduction and discussion section.

Comment 13

At the end of the introduction, there is no clear discussion of how this study differentiates from the previously mentioned studies, all of which use PBMCs. This lack of distinction and novelty among the studies is a significant concern. Additionally, the specific objective of this study is not clearly articulated in relation to the prior work. Clearly defining the study's objective is crucial, as it will guide the remainder of the work and must be novel to contribute significantly to scientific and clinical relevance.

Response

All previous studies determining telomere length in leukocytes from AMD patients have used the peripheral blood mononuclear cell (PBMC) fraction obtained after gradient centrifugation. PBMCs are a mixture of white blood cells, which limits the interpretation of the results. In our work, we isolated individual subpopulations (i.e., T lymphocytes, B lymphocytes, NK cells, or monocytes) and demonstrated for the first time the association of telomere length with different types of AMD. This is the novelty of the paper. Such results have never been published before.

All of the above is stated in the Introduction section.

Comment 14

The manuscript lacks details regarding the treatment regimen and the extent of antiangiogenic therapy administered to the patients.

Response

The article comprehensively conveys information about the treatment regimen in the M&M section: Patients in the latter two groups received anti-VEGF therapy for neovascular AMD (active macular neovascularization) and had a history of at least 20 injections in one eye. Patients undergoing antiangiogenic therapy received intravitreal injections of aflibercept 0.05 ml (2 mg) in the operating room using a 27 G needle, at least 3.5 mm from the limbus, following standard procedure. Three injections were performed at a four-week interval, followed by a treat-and-extend regimen.

Comment 15

I believe that the analysis conducted in the manuscript should be complemented with additional methods to verify the results. For instance, employing fluorescently labeled nucleic acid probes that specifically bind to telomeric repeats for Q-FISH could provide more detailed insights into telomere length by measuring fluorescence using flow cytometry. Alternatively, methods such as STELA could be used to detect shorter telomeres on specific chromosomes. Incorporating these techniques would strengthen the robustness of the findings and offer a more comprehensive assessment of telomere dynamics.

Response

In our work, we aimed to study telomere length in leukocytes using a commonly used validated technique, rather than to compare the validity of different methods. In our study, we used the qPCR method, the validity of which has been confirmed many times by comparison with other methods.

O’Callaghan, 2011

Cawthon, R.M., 2002

Gil, M.E, 2004

Comment 16

The Statistical Analysis section appears limited in detail. Could the authors expand significantly to provide a more thorough description of the statistical analyses employed, including the rationale behind the selection of methods? Additionally, it would be helpful to explain why a p-value of 0.008 was considered significant and to include details on effect size and estimated statistical power for the analyses performed.

Response

We thank the reviewer for this comment. The section that the reviewer mentioned is short due to the descriptive nature of all the statistical analysis that was performed. We were not really selecting the methods but rather used only those appropriate and common.

Throughout the manuscript the analysis only involved the comparison of pairs of quantitative parameters. Shapiro-Wilk test is the most used test to assess if the data follows a normal distribution. Not all the variables followed a normal distribution, therefore we used the Mann-Whitney U test to compare telomere lengths for each pair of groups, this is the only appropriate test in such a setting. With regards to the threshold of significance of 0.008, we apologize for possible lack of clarity. The Bonferroni correction approach which we used to account for multiple comparison suggests that a p-value threshold should be a ratio between conventional 0.05 and the number of comparisons (6 pairs of comparison groups in our case). Hence, 0.05/6=0.008 was used as a significance threshold. We have now clarified that better in the text.

With regards to the effect size and the statistical power, this was a retrospective study and therefore these parameters are not reported.

Comment 17

To ensure the effectiveness and purity of the cellular separations, the authors should have included a comparison of the values obtained using gradient centrifugation versus immunomagnetic separation. Providing this comparison would help validate the purity of the cell populations used in the study and ensure that the results are reliable.

Response

In our work, we aimed to study telomere length in leukocytes, but not to compare different cell separation protocols. In our study, we used routine immunomagnetic separation and confirmed the purity of the cells obtained by routine flow cytometry techniques. The purity in our work was more than 90% and this value was sufficient to obtain statistically different results in telomere length.

Comment 18

I am concerned that the current study does not present a significant novelty, as similar methodologies, such as magnetic separation of leukocytes for ocular diseases, have already been explored in previous publications. For instance, Vazirpanah et al. have conducted related research (https://journals.plos.org/plosone/article?id=10.1371/journal.pone.0176175 ). The study should emphasize how it differs from or builds upon these existing studies to demonstrate its unique contributions to the field.

Response

Again, the major novelty of our work is that we have demonstrated for the first time the association of telomere length in T lymphocytes, B lymphocytes, NK cells, and monocytes with different forms of age-related macular degeneration. We did not aim to develop a new method for cell separation or a new method for telomere detection (comment 15). In the paper mentioned by the reviewer, the authors worked with a different disease, avian uveitis, and the comparison with AMD is not correct. In addition, the authors demonstrated opposite dynamics in telomere length between some leukocyte subpopulations.

Comment 19

I have noted inconsistencies between the p-values reported in the tables and those described in the Materials and Methods section. For instance, in Table S1, the p-value for NK cells versus T cells is reported as 0.006, while the Materials and Methods section states that a p-value of 0.008 is considered significant. Additionally, similar discrepancies are observed in Tables S2 and S5, where p-values are reported as being less than 0.008 but greater than 0.001. The authors should address these contradictions by clarifying the criteria used for significance and ensuring consistency across all reported values.

Response

There is no mistake here. In accordance with the Bonferroni correction method the p<0.008 (see the answer for comment 16) was considered and significant. Therefore, any p-values in our tables such as 0.006 or anything between 0.008 and 0.001 are significant as they are all <0.008. Now that we have clarified the rationale behind the value of 0.008, we hope this should become more transparent.

Comment 20

In Table 1, the term "IQR" is used but not defined.

Response

IQR is defined in Statistics subsection 2.4.

Comment 21

The study would benefit from a discussion on the concept that the eye is an immunoprivileged organ, isolated from the rest of the body by the blood-retinal barrier. The authors should explain the relevance of studying peripheral blood cells in the context of an ocular disease that primarily affects the retina, which has this controlled barrier. How do the findings in peripheral blood cells relate to the retinal pathology of age-related macular degeneration, given the isolation and unique immune environment of the eye?

Response

Thank you for this concern. The retina is protected from the bloodstream by a blood-retinal barrier, preventing uncontrolled substance diffusion between blood and the retina. The loss of barrier in-tegrity is directly linked to neovascularization, stemming from the absence of the char-acteristic blood-brain barrier structure in the new vessels observed in AMD [65].

[3] Apple, D.J.; Goldberg, M.F.; Wyhinny, G. Histopathology and ultrastructure of the argon laser lesion in human retinal and choroidal vasculatures. Am. J. Ophthalmol. 1973, 75, 595–609, doi:10.1016/0002-9394(73)90812-x

Once again, we thank the reviewer and the editorial office for thorough review and hope that the corrections have improved this manuscript.

Reviewer 3 Report

Comments and Suggestions for Authors

In this manuscript, Anait S. Khalatyan et al. summarize the relationship between leukocytes and telomere length in AMD patients and healthy individuals, revealing that telomere length in monocytes has the strongest association with AMD. Overall, this review is valuable for clinical medicine research, suggesting that telomere length in monocytes may become a diagnostic hallmark of AMD.

The manuscript also describes the method for isolating leukocytes and the efficiency of the purification process. If telomere length in monocytes could become a diagnostic hallmark of AMD, what is the frequency of monocytes in AMD patients? Could the author provide the frequencies of T, B, NK, and monocytes in patients PBMC?

Additionally, there are some labeling errors in the manuscript. On page 5, Fig. 1 A-C should be corrected to A-D, and the same applies to the Figure 1 legend. In the penultimate paragraph of the discussion section, Fig. 3C should be corrected to Fig. 3D. Please review all the figures and legends carefully.

Comments on the Quality of English Language

The quality of english language is pretty good.

Author Response

We appreciate the reviewer’s responses, including characterizing the paper as “valuable for clinical medicine research”. In the following response, we address each calling for changes, indicating where relevant corrections have been added to the body of the manuscript and their locations. For reviewers’ convenience, all corrections are tracked in red color in the main text file.

We thank all reviewers for their comments and suggestions.

Response to the comments from the Reviewer #3

Comment 1

The manuscript also describes the method for isolating leukocytes and the efficiency of the purification process. If telomere length in monocytes could become a diagnostic hallmark of AMD, what is the frequency of monocytes in AMD patients? Could the author provide the frequencies of T, B, NK, and monocytes in patients PBMC?

Response:

Thank you for this concern. Unfortunately, we did not look at the frequencies of leukocyte subpopulations in our work, and this would need to be a separate study.

To answer this concern, we have found several articles on related topics and discussed them in the discussion section.

Comment 2

Additionally, there are some labeling errors in the manuscript. On page 5, Fig. 1 A-C should be corrected to A-D, and the same applies to the Figure 1 legend. In the penultimate paragraph of the discussion section, Fig. 3C should be corrected to Fig. 3D. Please review all the figures and legends carefully.

Response

Thank you for pointing out these mistakes. We have corrected them.

Once again, we thank the reviewer and the editorial office for thorough review and hope that the corrections have improved this manuscript.
